# Immunorthodontics: PD-L1, a Novel Immunomodulator in Cementoblasts, Is Regulated by HIF-1α under Hypoxia

**DOI:** 10.3390/cells11152350

**Published:** 2022-07-30

**Authors:** Jiawen Yong, Sabine Gröger, Julia von Bremen, Joerg Meyle, Sabine Ruf

**Affiliations:** 1Department of Orthodontics, Faculty of Medicine, Justus Liebig University Giessen, 35392 Giessen, Germany; sabine.e.groeger@dentist.med.uni-giessen.de (S.G.); julia.v.bremen@dentist.med.uni-giessen.de (J.v.B.); sabine.ruf@dentist.med.uni-giessen.de (S.R.); 2Department of Periodontology, Faculty of Medicine, Justus Liebig University Giessen, 35392 Giessen, Germany; joerg.meyle@dentist.med.uni-giessen.de; 3Stomatology Hospital, School of Stomatology, Zhejiang University School of Medicine, Zhejiang Provincial Clinical Research Center for Oral Diseases, Key Laboratory of Oral Biomedical Research of Zhejiang Province, Cancer Center of Zhejiang University, Hangzhou 310003, China

**Keywords:** cementoblasts, hypoxia, PD-L1, HIF-1α, orthodontic, immunorthodontics

## Abstract

Recent studies have revealed that hypoxia alters the PD-L1 expression in periodontal cells. HIF-1α is a key regulator for PD-L1. As hypoxia presents a hallmark of an orthodontically induced microenvironment, hypoxic stimulation of PD-L1 expression may play vital roles in immunorthodontics and orthodontically induced inflammatory root resorption (OIIRR). This study aims to investigate the hypoxic regulation of PD-L1 in cementoblasts, and its interaction with hypoxia-induced HIF-1α expression. The cementoblast (OCCM-30) cells (M. Somerman, NIH, NIDCR, Bethesda, Maryland) were cultured in the presence and absence of cobalt (II) chloride (CoCl_2_). Protein expression of PD-L1 and HIF-1α as well as their gene expression were evaluated by Western blotting and RT-qPCR. Immunofluorescence was applied to visualize the localization of the proteins within cells. The HIF-1α inhibitor (HY-111387, MedChemExpress) was added, and CRISPR/Cas9 plasmid targeting HIF-1α was transferred for further investigation by flow cytometry analysis. Under hypoxic conditions, cementoblasts undergo an up-regulation of PD-L1 expression at protein and mRNA levels. Silencing of HIF-1α using CRISPR/Cas9 indicated a major positive correlation with HIF-1α in regulating PD-L1 expression. Taken together, these findings show the influence of hypoxia on PD-L1 expression is modulated in a HIF-1α dependent manner. The HIF-1α/PD-L1 pathway may play a role in the immune response of cementoblasts. Thus, combined HIF-1α/PD-L1 inhibition could be of possible therapeutic relevance for OIIRR prevention.

## 1. Introduction

Orthodontic tooth movement (OTM) takes place as a direct consequence of connective tissue remodeling within the periodontium which occurs through a localized inflammatory response induced by external mechanical forces [1] when using removable or fixed orthodontic appliances [2]. The compressive force results in an altered homeostasis of the periodontal ligament (PDL) cells and the host immune response [3]. This orthodontic force causes capillary vasodilatation within the blood vessels of the PDL, in turn resulting in migration of immune cells and the expression of various cytokines such as GAS-6 [4]. These soluble cytokines regulate the process of cementum resorption, so-called orthodontic-induced inflammatory root resorption (OIIRR), in response to orthodontic force [5]. Cementum seems to be able to protect the tooth root from OIIRR [6]. PDL cells (mainly cementoblasts) are capable of repairing resorption pits with new cementum [7,8] and play an immunomodulatory role under hypoxic conditions [9].

The programmed cell death receptor ligand 1 (PD-L1), also known as B7 homolog 1 (B7-H1) or cluster of differentiation (CD) 274, plays a crucial role in preventing tissue damages due to excessive immune responses in the inflammatory tissues [10,11]. This immune checkpoint is a regulator of immune activation. It is expressed on antigen-presenting cells such as monocytes, macrophages [12], dendritic cells (DCs) [13] and B cells and is upregulated upon activation. Until now, besides in different human carcinomas, PD-L1 has also been found to be expressed on OIIRR-associated periodontal tissue cell types including human osteoblasts [14], which have a similar phenotype as cementoblasts, human osteoclasts [15], human periodontal ligament cells [16], human gingival fibroblasts [17] and human gingival keratinocytes [18]. It is suggested that PD-L1 plays a pivotal role in delivering an inhibitory signal to programmed cell death 1 (PD-1) expressing T cells, resulting in an immune system impairment [19]. Until now, the expression of PD-L1 on cementoblasts has not been verified. 

During OTM the concomitant compressive forces on blood vessels create a hypoxic microenvironment [6] by reducing the local oxygen supply within the PDL in which cementoblasts are exposed to locally decreased cellular oxygen tension, so-called hypoxia [20]. Cobalt (II) chloride (CoCl_2_) [21] has previously been used as a hypoxia mimicking agent and it has been proven to effectively induce the HIF-1α expression stability on cementoblasts in vitro [22]. Furthermore, Noman et al. (2014) showed that CoCl_2_-induced hypoxia induces a PD-L1 expression directly through HIF-1α in myeloid-derived suppressor cells (MDSCs) and its blockade under hypoxia promoted MDSC-mediated T cell activation [23]. Kyrkanides et al. (2021) reported that HIF-1α may be the first host response in OTM [24]. Very recent in vivo investigations by Kirschneck et al. (2021) demonstrated that HIF-1α from myeloid cells participates in the regulation of OIIRR. After orthodontic treatment, mice lacking HIF-1α in myeloid cells had an accelerated OTM compared to wildtype, suggesting a bone-protective activity of HIF-1α during OTM [25]. Another study also showed that the mechanotransducive stabilization of HIF-1α in PDL fibroblasts occurs under compressive forces, but not tensile forces and is regulated by phosphorylation of ERK [26]. This evidence corresponds to the fact that the activation of transcription factor HIF-1α resulting from hypoxia or orthodontic forces plays regulatory roles in the periodontium during OIIRR. 

OIIRR is associated with interactions between an inflammatory microenvironment and the host immune response. Although an involvement of PD-L1 in the progression of periodontitis has been suggested, the functional contribution of PD-L1 expressed in OIIRR has not been investigated. Moreover, CoCl_2_-mimicked hypoxia has been reported to correlate with the immunomodulatory effects of PD-L1, effects that are dependent on target genes regulated by HIF-1α [23]. Considering that PD-L1 is up-regulated in various non-immune tissues as well as immune cells under inflammatory conditions, in this study, we aim to explore the expression and correlation of PD-L1/HIF-1α in cementoblasts under CoCl_2_-mimicked hypoxia.

## 2. Materials and Methods

### 2.1. Cell Culture 

Immortalized murine mouse cementoblast (OCCM-30) cell line [27] was kindly provided by Prof. J. Deschner and Dr. M. Nokhbehsaim (Department of Periodontology, University of Bonn, Germany). For the experiment, OCCM-30 cells were cultured in a complete growth medium (α-minimal essential medium, α-MEM, 11095-080, Gibco, Gaithersburg, MA, USA) containing 10% Fetal Bovine Serum (FBS) (10270-106, Gibco) and 1% Penicillin/Streptomycin (15140-122, Gibco) in 6-well cell culture plates (657160, Greiner bio-one, Frickenhausen, Germany) until proliferatory outgrowth of adherently growing cementoblasts was observed. The cells used in the study were from the 3rd to 5th passage and were seeded at a density of 1 × 10^6^ cell/well under standard cell incubation conditions (5% CO_2_, 37 °C, water saturated).

### 2.2. Hypoxic Conditioning of Cementoblasts

For hypoxic experiments, OCCM-30 cells were cultivated supplemented with 100 μM or 400 μM cobalt (II) chloride hexahydrate (CoCl_2_) for indicated time periods to mimick the different hypoxic culture conditions [28]. Briefly, CoCl_2_ (C8661, Sigma-Aldrich, Taufirchen, Germany) was dissolved directly to the growth medium and sterilized through a sterile 0.2 μm spare membrane filter (Z333905-1EA, Merck, Darmstadt, Germany) to reach final concentrations of 100 μM or 400 μM. These concentrations are based on the hypoxic concentration established in previous studies [22,28]. Cells cultured without CoCl_2_ served as the normoxic control. 

### 2.3. Pharmacological Inhibitor

For the HIF-1α inhibition experiments, 0, 5 nM (1.85 μg/mL), 10 nM (3.7 μg/mL), 20 nM (7.7 μg/mL) HIF-1α inhibitor (IDF-11774) (HY-111387, MedChemExpress, Monmouth Junction, NJ, USA) were used. The same amount of DimethyIsulfoxide (DMSO) (D8418-50ML, Sigma-Aldrich) was used as a negative-control group.

### 2.4. CRISPR/Cas9 Plasmid Transfection

The cells were modified using the CRISPR/Cas9 technique and the HIF-1α gene was knocked down according to the manufacturer’s protocol [29]. Twenty-four hours prior to transfection, 2 × 10^5^ cells/well was seeded in 3 mL of antibiotic-free standard growth medium. After reaching a 50% confluent growing of healthy cells, the CRISPR/Cas9 KO Plasmid transfection was performed according to the recommended procedures. For the plasmid DNA solution, HIF-1α CRISPR/Cas9 KO Plasmid (m2) (sc-420856-KO-2, K1617, Santa Cruz Biotechnology, Santa Cruz, CA, USA) of 2 μg was added to 140 μL serum/antibiotic-free medium (sc-108062, Santa Cruz, Bad Homburg, Germany) and incubated at room temperature (RT) for 5 min. For the transfection solution, UltraCruz^@^ Transfection Reagent (sc-395739, Santa Cruz) of 10 μL was added to 140 μL serum/antibiotic-free medium (sc-108062, Santa Cruz) and incubated at RT for 5 min. Then, the plasmid DNA solution was directly added dropwise into the transfection solution to obtain about 300 µL transfection complex and incubated at RT for 15 min. Afterwards, the cells were incubated with the transfection complex solution at 37 °C for 48 h and then were re-incubated in standard growth medium at 37 °C for an additional 12 h. The single transfection solution addition is provided as a negative control. The efficiency of transfection was verified by RT-qPCR and visually confirmed by detection of the green fluorescent protein (GFP) via GFP antibody (sc-9996, Santa Cruz) (dilution 1:200).

For the experiment, cells were subsequently stimulated in 100 μM or 400 μM CoCl_2_-induced hypoxia for 24 h, and the target genes/proteins were detected by Western blot (WB) and/or reverse transcription–quantitative real time polymerase chain reaction (RT-qPCR) analysis.

### 2.5. RNA Isolation and Quality Measurement

For RNA isolation, the treated cells were first washed with 1 mL phosphate-buffered saline (PBS) (10010023, Gibco) per well and then harvested with 350 μL buffer RLT (74104, Qiagen, Germany) supplemented with 1% β-mercaptoethanol (β-ME). Afterwards, the RNA was isolated with an RNase Mini Kit (74104, Qiagen, Hilden, Germany) following the producer’s protocol including an on-column DNA digestion (RNase-Free DNase, Qiagen, Germany) and a DNase step for removal of genomic DNA. In order to control the success of the purification and to ensure a uniform cDNA synthesis, every sample was measured twice (Nanodrop 2000, Thermo Fisher Scientific, Waltham, MA, USA). 

After isolation, according to the manufacturer’s instructions, the eluted RNA purity and quantity of each sample was verified photometrically by optical density (OD) readings of the A260/280 nm ratio (Nanodrop 2000, Thermo Fisher Scientific, USA). The spectrophotometric ratio of A260/280 varied >1.80 and A260/230 values yield a ratio >2.0. Thus, the isolated RNA could be regarded as free of protein and polyphenols/polysaccharides contaminants.

### 2.6. Reverse Transcription and RT-qPCR Analysis

For the cDNA synthesis, 1 μg of the RNA/sample was transcribed into cDNA by using the iScript^TM^ cDNA Synthesis Kit (1708891, Bio-Rad, Feldkirchen, Germany). All steps from RNA isolation to cDNA synthesis were performed parallel for all samples of every experiment in order to avoid experimental variations. 

For every RT-qPCR amplification, 8 μL DNase-free water (AM9935, Sigma-Aldrich), 10 μL SsoAdvanced^TM^ Universal SYBR Green Supermix (1725270, Bio-Rad), 1.0 μL cDNA and 1.0 μL primer were mixed and then added 1.0 μL target primer to bring the final 20 μL volume reaction. Primers were designed by Bio-Rad including *CD274* (*PD-L1*, qMmuCED0044192) and *HIF-1α* (qMmuCID0005501). For the normalization of target genes (Rel. mRNA), reference genes *PPIB* (qMmuCED0047854), which have been shown to be stably expressed in OCCM-30, were used [30]. The RT-qPCR was performed using the following protocol: 95 °C/30 s followed by 40 cycles of 95 °C/15 s and 60 °C/30 s. The melting curves after each RT-qPCR cycles that were carried out between 50 °C and 95 °C with a plate read every 0.5 °C increment after holding the temperature for 5 s with continuous fluorescence acquisition. Detection was performed using the CFX96^TM^ Real-Time System (C1000^TM^ Thermal Cycler, Bio-Rad). 

The relative gene expression used for the statistical analysis was calculated as 2^−ΔΔCq^ compared to the normoxic control group to set the relative gene expression to 1.0. Results were analyzed using the Bio-Rad CFX Manager software (version 3.1, Bio-Rad, Feldkirchen, Germany). All cDNA samples were tested as three replicates per reference gene.

### 2.7. Immunofluorescence (IF)

The HIF-1α and PD-L1 expression within OCCM-30 cells in response to hypoxic conditions were detected by IF. Cells were cultured on sterile Falcon™ Chambered Cell Culture Slides (354108, Thermo Fisher Scientific) until 50% confluence. Afterwards, they were fixed with 4% paraformaldehyde (pH 7.4, 158127, Sigma-Aldrich) for 10 min and permeabilized with 0.5% Triton™ X-100 Surfact-Amps™ Detergent Solution (28313, Thermo Fisher Scientific) for 20 min at RT. Then, cells were kept in blocking buffer (#12411, Cell Signaling Technology, Frankfurt a. Main, Germany) for 30 min at RT and washed twice with 1 × PBS with 0.02% Tween-20 (PBST) (P1379, Sigma-Aldrich) for each step. Afterwards, the cells were incubated with PD-L1 polyclonal antibody (PA5-20343, Invitrogen, Leipzig, Germany) (dilution 1:250) or HIF-1α polyclonal antibody (PA1-16601, Invitrogen) (dilution 1:400) at 4 °C overnight. After washing three times with 1 × PBST for 5 min, the cells were incubated with DyLight^@^ 488 goat anti-rabbit polyclonal secondary antibody (ab96899, Abcam, Cambridge, UK) (dilution 1:500), which conjugated to fluorescein isothiocyanate for 1 h in the dark at RT. After washing with 1 × PBST, DNA was stained using a fluorescent Mounting Medium with 4′,6-diamidino-2-phenylindole (DAPI) (ab104139, Abcam) for 5 min. Slides were imaged using a high-resolution fluorescence microscope (Leica Microsystems, Wetzlar, Germany) and photographed.

### 2.8. Flow Cytometry Analysis

For the FACS analysis, both floating and adherent cells were harvested with Gibco™ StemPro Accutase Cell Dissociation Reagent (A1110501, Gibco) and collected. The samples were washed twice with FACS buffer (554657, BD Pharmingen, Heidelberg, Germany) and adjusted to a concentration of 1 × 10^6^ cells/mL in pre-cooled FACS buffer and transferred to polystyrene round-bottom 12 × 75 mm BD Falcon tubes (10579511, Thermo Fisher Scientific) on ice. The control groups used for compensation and quadrants were set up with isotype antibody (ab171870, abcam)-stained cells. For the experimental groups, cells were incubated and stained with PD-L1 polyclonal antibody (PA5-20343, Invitrogen) (dilution 1:250) or HIF-1α polyclonal antibody (PA1-16601, Invitrogen) (dilution 1:400) in FACS buffer. Following incubation for 1 h at RT in the dark, 400 μL of DyLight^@^ 488 goat anti-rabbit polyclonal secondary antibody (ab96899, Abcam) (dilution 1:500) was added to each tube. Finally, both cell surface and intracellular PD-L1 or HIF-1α staining were performed to detect immediate changes and cells were acquired by a FACS Vantage Flow Cytometer (SP6800 Spectral Analyzer, Sony Biotechnology, Berlin, Germany) within 1 h. Data were obtained from SP6800 Spectral Analyzer software (version 2.0.2, Sony corporation, Berlin, Germany) and analyzed by FlowJo software (version 10, Treestar, California, CA, USA). For every sample, 1 × 10^4^ events were recorded, and this assay was performed in triplicate. 

### 2.9. Protein Extraction and Western Blot Analysis

The cells were collected in Pierce^TM^ RIPA buffer (89901, Thermo Fisher Scientific) supplemented with 3% phosphatase and protease inhibitors (78442, Thermo Fisher Scientific) on ice and incubated for 30 min. Protein concentration was measured using the Pierce^TM^ BCA Protein Assay Kit (23225, Thermo Fisher Scientific) on a Nanodrop 2000 Spectrophotometer. Protein lysates (20 μg) were separated by electrophoresis on 4-20% Mini-PROTEAN^@^ TGX^TM^ Precast Gel (#4561093, Bio-Rad) and transferred to a nitrocellulose membrane (1704271, Bio-Rad) using the Trans-Blot Turbo Transfer System (Bio-Rad). 

Membranes were blocked with 5% non-fat milk solution (T145.1, ROTH) and then incubated for 1 h at RT employing either PD-L1 polyclonal antibody (PA5-20343, Invitrogen) (dilution 1:1000) or HIF-1α polyclonal antibody (PA1-16601, Invitrogen) (dilution 1:1000). β-actin (ab8227, Abcam) (dilution 1:2000) was used to standardize the loading. Polyclonal Goat Anti-Rabbit (P0448, Dako) HRP at a dilution 1:2000 was used as secondary antibody. The signals were detected utilizing Amersham ECL Western Blotting Detection Reagents (9838243, GE Healthcare, Freiburg, Germany) on Amersham Hyperfilm (28906836, GE Healthcare) on OPTIMAX X-Ray Film Processor (11701-9806-3716, PROTEC GmbH, Oberstenfeld, Germany). The ImageJ program (version 2, NIH, Washington, DC, USA) was used for densitometric analysis.

### 2.10. Statistical Analysis

Statistical analyses were performed using GraphPad Prism 8.0 software (GraphPad software Inc., San Diego, CA, USA). All values are expressed as means ± standard deviation (SD) and analyzed using Student’s *t*-test for unpaired samples to determine the statistically significant differences between two groups. A one-way ANOVA was used for multiple comparisons involving more than two groups. Differences were considered statistically significant at a *p* value of < 0.05. Data distribution was analyzed using the Kolmogorov–Smirnov and the Shapiro–Wilk test and visually using QQ plots. Each experiment was performed in triplicate and repeated successfully at least three times.

## 3. Results

### 3.1. Up-Regulation of PD-L1 in Cementoblasts under Hypoxic Conditions

The results show that PD-L1 expression was substantially increased at mRNA- and protein level when treated with either 100 or 400 μM hypoxic conditions during 12 and 24 h as compared to normoxia (Figure 1A,B,E,F). Similar results were observed visually through the immunofluorescent staining of PD-L1, as it was highly up-regulated in both hypoxic conditions, whereas under normoxia, only a slight expression was detected (Figure 1C). As shown in Figure 1D, hypoxia significantly increased the percentage of PD-L1^+^ cementoblasts (Figure 1D).

Thus, it appears as if the CoCl_2_-mimicked hypoxia significantly up-regulates the PD-L1 expression in a dose- and time-dependent manner at both the mRNA- and protein levels in cementoblasts.

### 3.2. Increased Expression Levels of HIF-1α in Cementoblasts in Response to Hypoxia

We further investigated whether hypoxia can induce HIF-1α expression in OCCM-30 cells. RT-qPCR analysis demonstrated that simultaneous with the PD-L1 induction, HIF-1α was up-regulated in cementoblasts under hypoxia (Figure 2A,B). Furthermore, hypoxia—especially under 400 μM conditions—significantly increased the protein expression level of HIF-1α during 8 to 24 h (Figure 2C,D). Accordingly, the IF staining revealed that the HIF-1α expression was significantly increased during 24 h in response to hypoxia (Figure 2E). Therefore, 400 μM hypoxic stimulation was selected for the next steps.

### 3.3. Knockdown of HIF-1α Expression in Cementoblasts

To dissect the roles of the HIF-1α in PD-L1 up-regulation under hypoxia, the OCCM-30 cells were treated with different kinetics and concentrations of a pharmacological inhibitor of HIF-1α (IDF-11774) or the control (DMSO). As shown in Figure 3A,B, on selective inhibition of HIF-1α, a significant inhibition in the degree of hypoxia-induced HIF-1α expression was demonstrated in cementoblasts co-cultivated with IDF-11774 (20 nM). Under hypoxic conditions, the IF staining showed that the expression of HIF-1α in cells was strongly inhibited compared to the control (Figure 3C,D). Moreover, a CRISPR/Cas9-mediated knockdown of HIF-1α under hypoxia significantly decreased mRNA expression of HIF-1α in OCCM-30 cells (Figure 3E).

These findings confirm that the effective HIF-1α blockade/knockdown under hypoxia were due to HIF-1α inhibition by IDF-11774 at a concentration of 20 nM. 

### 3.4. The Increased Expression of PD-L1 Is Directly Regulated by HIF-1α under Hypoxia in Cementoblasts

To explore whether the PD-L1 expression response to hypoxia in cementoblasts was related with HIF-1α, the effective IDF-11774 or CRISPR/Cas9 plasmid was used. The WB results showed that a blockade of HIF-1α under hypoxia significantly abrogated the up-regulation of PD-L1 protein under 400 μM hypoxic conditions (Figure 4A,B,D). Even under a 100 μM hypoxia, the PD-L1 protein was inhibited as a result of the HIF-1α inhibition (Figure 4C). Interestingly, after a 24 h co-stimulation with IDF-11774, it was observed that the amount of living cells was decreasing, and the apoptosis of cells was induced under both hypoxic conditions (Figure 4C,D). Under 400 μM hypoxia, IDF-11774 decreased the percentage of PD-L1^+^ cells, whereas the percentage of PD-L1^+^ cells in 100 μM group was slightly decreased but without a significant difference (Figure 4E). Furthermore, the HIF-1α gene knockdown showed down-regulatory effects on PD-L1 gene expression in response to both hypoxic conditions (Figure 4F). Thus, the results suggest that HIF-1α is a major regulator of PD-L1 mRNA and protein expression.

## 4. Discussion

In the results of the present study indicate that hypoxic conditions significantly augment the PD-L1 expression on cementoblasts during the long exposure stages. Furthermore, PD-L1 is found to interact with HIF-1α. It furthermore acts as a novel target on cementoblasts under hypoxia. Since hypoxia is one of the major features of an orthodontic-induced microenvironment, the HIF-1α/PD-L1 pathway is hypothesized to be the major modulator in immunorthodontics during OTM.

As shown previously by Keir et al. (2008), PD-L1 is constitutively expressed on a wide variety of cells [31] such as B cells, T cells, macrophages and DCs and is up-regulated upon their activation [32]. In the field of periodontology, PD-L1 is reported to be expressed in oral carcinoma cells [33,34] and in human gingival keratinocytes [18]. It plays important roles in a co-opted and maladaptive immune shield that protects the cells from the inflammatory microenvironment [18]. Evidence shows that the increased expression of PD-L1 has regulatory effects on the T cells activation and maintains the cell hemeostasis [35]. Our results clearly show that hypoxia is a novel critical modulator of PD-L1 expression on cementoblasts in vitro. It would be of major interest to study the potential contribution of an increased hypoxia-induced PD-L1 on cementoblasts, interacting with antigen-presenting cells in regulating the immune suppression within the hypoxic microenvironment. 

The kinetics analysis of hypoxia-induced PD-L1 expression is similar to that observed upon the HIF-1α expression. Moreover, the present results show that PD-L1 up-regulation is counteracted by the HIF-1α inhibition, indicating that PD-L1 is the target, and it depends on HIF-1α under hypoxic conditions. It is important to underline that the present results are in agreement with previous findings [36]. Duijn et al. (2021) reported a pronounced influence of hypoxia on IFNγ-induced *PD-L1* mRNA expression, which is controlled by the HIF1 knockdown as well as at a 952 bp *PD-L1* promoter fragment. Their findings indicate a role of hypoxia on PD-L1 expression [36]. However, Ruf et al. (2016) showed that in renal cell carcinoma PD-L1 is regulated by HIF-2α, not HIF-1α [37]. Both, HIF-1 and HIF-2 are key mediators of cellular response to hypoxia and are expressed in osteoblasts [38]. Therefore, the expression of HIF α subunits (HIF-1α, HIF-2α, HIF-3α) and β subunits (Arnt and ARNT2) to clarify the hypoxia-inducible factors pattern in cementoblasts in response to hypoxic conditions should further be investigated.

The functional profile of PD-L1 in a human inflammatory environment has previously been reviewed with regard to the regulation of the T cell immune response. T cells are required in the immune response and play essential roles in OTM and OIIRR [39]. Previous data showed that overexpressed PD-L1 in the orthodontic-induced microenvironment engages PD-1 and subsequently triggers inhibitory downstream signaling of the T cell receptor [40]. Under healthy conditions, the PD-1/PD-L1 pathway maintains host immune homeostasis, while under inflammatory conditions, interactions between PD-1 and its ligand PD-L1 serve to inhibit T cell responses, which protects the lesion from hyperactivated T cells in cancer [37]. It was reported that the overexpression of PD-L1 in gingival basal keratinocytes in PD-L1 generated K14/PD-L1 transgenic mice reduces the periodontal inflammation and alveolar bone resorption in a ligature-induced periodontitis model [41]. Here, our results indicate that cementoblasts can express PD-L1 upon hypoxic stimulation and that HIF-1α stabilized by hypoxia was able to activate the expression of PD-L1 in cementoblasts (Figure 1).

However, PD-L2, which is restricted to expressed on antigen-presenting cells, is another ligand with stronger binding affinity to PD-1 in comparison to PD-L1 [42]. Engagement of PD-1 with either of its two membrane-bound ligands, PD-L1 or PD-L2, can inhibit immune responses [42]. Thus, the expression and roles of PD-L2 in cementoblasts under hypoxic conditions should be further studied. 

Different hypoxic induction sets have been introduced by means of physical influences (hypoxia chamber) or chemical agents (nickel chloride and CoCl_2_); the former is difficult to maintain due to local steady oxygen tension, while the latter, CoCl_2_, was used effectively to induce stable HIF-1α expression in order to investigate the signaling interaction network with HIF-1α. Further investigations should also be performed using different hypoxic conditions to verify the mechanisms raised in the present study. Moreover, this mechanism should be proven by HIF-1α gene knock-out mice because of the incomplete knockdown of HIF-1α by the CRISPR/Cas9 technique in the cell line.

We therefore hypothesize that aberrant immune checkpoint PD-L1 expression facilitates an antigenic escape from the immune surveillance, making the ligand a potential anti-OIIRR target [37]. In the present study, we identified a mechanism underlying the role of PD-L1/HIF-1α in cementoblasts. Furthermore, PD-L1 up-regulation was found to be HIF-1α dependent. Thus, combined PD-L1/HIF-1α inhibition could be of possible therapeutic relevance for OIIRR in orthodontic therapy.

## 5. Conclusions

We demonstrated that hypoxia up-regulated PD-L1 on cementoblasts via HIF-1α. A blockade of HIF-1α under hypoxia abrogated the immune response of cementoblasts. For the first time, these results show a link between the immune checkpoint PD-L1, hypoxia-induced HIF-1α and cementoblasts immune suppression under an orthodontic-induced microenvironment. Therefore, a combination therapy targeting OIIRR by using HIF-1α along with PD-L1 inhibition may be beneficial for boosting the immune response in OTM.

## Data Availability

The datasets used and/or analyzed during the current study are available from the corresponding author on reasonable request.

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
