# Peer review of "Immunorthodontics: PD-L1, a Novel Immunomodulator in Cementoblasts, Is Regulated by HIF-1α under Hypoxia"

_cells, 2022, doi:10.3390/cells11152350_

Round 1
Reviewer 1 Report
The topic of the article is very specific and difficult to explain but I think that the introduction could be improved with an enlarged explanation of the role of HIF-1alpha in tooth movement.
Reviewer 2 Report
In this report, Yong et al. found that HIF-1α under Hypoxia significantly up-regulates the PD-L1 expression in cementoblasts. They found CoCl2-mimicked hypoxia-induced PD-L1 expression in a dose- and time-dependent manner at both the mRNA- and protein levels. HIF-1α Inhibitor or knockdown under hypoxia suppressed PD-L1 expression in cementoblasts. It has also been reported that Hypoxia also upregulates the expression of PD-L1 in malignant and immune regulatory cells. Although the current study provides in vitro evidence that HIF1a is required for PD-L1 expression in cementoblast cells, the above-mentioned studies undercut the novelty of the current study.
1. Does Hypoxia regulate PD-L2 expression in cementoblasts?
2. It might be good to confirm the findings using another hypoxic condition.
3. Page 4 line 254, protein expression level of HIF-1α during 8 to 24 hours was shown in Figure 2 C-D, but not Figure 3 C-D.
4. The knockdown efficiency of CRIPR/Cas9 is less than 50% in Figure 3E, the authors should use more CRIPR/Cas9 plasmids and avoid off-target effects.
5. The authors need to indicate how many independent experiments to get the summary data in figure legends such as Figure 1E,F, Figure 3E.
6. The high dose DMSO down-regulates the expression of HIF-1a in Figure 3A, the authors should examine the cell viability.
7. How about HIF-2a expression under hypoxic condition?
Round 2
Reviewer 2 Report
The authors addressed all the points I had raised.